# Photodegradation of Ciprofloxacin, Clarithromycin and Trimethoprim: Influence of pH and Humic Acids

**DOI:** 10.3390/molecules26113080

**Published:** 2021-05-21

**Authors:** Lucía Rodríguez-López, Raquel Cela-Dablanca, Avelino Núñez-Delgado, Esperanza Álvarez-Rodríguez, David Fernández-Calviño, Manuel Arias-Estévez

**Affiliations:** 1Soil Science and Agricultural Chemistry, Faculty of Sciences, University of Vigo, 32004 Ourense, Spain; lucia.rodriguez.lopez@uvigo.es (L.R.-L.); mastevez@uvigo.es (M.A.-E.); 2Department Soil Science and Agricultural Chemistry, Engineering Polytechnic School, University of Santiago de Compostela, 27002 Lugo, Spain; raquel.dablanca@usc.es (R.C.-D.); esperanza.alvarez@usc.es (E.Á.-R.)

**Keywords:** antibiotics, degradation, electrochemical environment, emerging pollutants, organic mater

## Abstract

In view of the rising relevance of emerging pollutants in the environment, this work studies the photodegradation of three antibiotics, evaluating the effects of the pH of the medium and the concentration of dissolved organic matter. Simulated light (with a spectrum similar to that of natural sunlight) was applied to the antibiotics Ciprofloxacin (Cip), Clarithromycin (Cla) and Trimethoprim (Tri), at three different pH, and in the presence of different concentrations of humic acids. The sensitivity to light followed the sequence: Cip > Cla > Tri, which was inverse for the half-life (Tri > Cla > Cip). As the pH increased, the half-life generally decreased, except for Cla. Regarding the kinetic constant k, in the case of Cip and Tri it increased with the rise of pH, while decreased for Cla. The results corresponding to total organic carbon (TOC) indicate that the complete mineralization of the antibiotics was not achieved. The effect of humic acids was not marked, slightly increasing the degradation of Cip, and slightly decreasing it for Tri, while no effect was detected for Cla. These results may be relevant in terms of understanding the evolution of these antibiotics, especially when they reach different environmental compartments and receive sunlight radiation.

## 1. Introduction

In recent years, antibiotics have been used massively to combat bacterial diseases, both in humans and in domestic animals, being recognized as emerging pollutants. This has even increased their scientific interest, as well as the concern of public administrations regarding this topic [1]. To be noted that, once administered, a significant percentage of antibiotics for human use are excreted through feces and urine, entering wastewater [2]. Furthermore, antibiotics are generally insufficiently eliminated in wastewater treatment plants [3,4,5], sometimes reaching relevant concentrations in waterways receiving effluents. But, in addition, its presence in waterbodies can also be related to aquaculture and agriculture activities [6], since antibiotics are abundantly used to prevent infections, and even in some countries they are also used as growth promoters, both for mammals and fish farming [7]. In fact, various antibiotics have been detected in surface waters in recent years [6,8,9,10]. These environmental problems, with potential repercussions on public health, are fundamentally related to toxic effects on different living organisms and to the development of resistance to these therapeutic molecules [11].

Degradation of the antibiotics that reach waterbodies is of main importance to avoid/diminish the magnitude of potential environmental issues. Within degradation processes, it is worth highlighting biodegradation and abiotic degradation, with especial relevance of photodegradation for the latter. As indicated by different authors [12,13], it is unlikely that biodegradation reach quantitative importance in waterbodies, with which photodegradation acquires greater prevalence in this specific environment.

The evaluation of photodegradation processes can provide information about the persistence of antibiotics in the environment when they receive sunlight, but also about their possible removal during wastewater treatment [14,15]. It is relevant that the photodegradation process can be affected by physicochemical conditions, such as pH, and by the presence of soluble organic matter. In fact, the effect of pH has been studied for some types of antibiotics, such as tetracyclines [16] and sulfonamides [17], finding increases in degradation for both types as a function of increasing pH. The effect of soluble organic matter has also been studied for some antibiotics, such as sulfadiazine, showing that its photodegradation increased with increasing concentrations of soluble organic matter [17,18]. However, the effects of pH and soluble organic matter have not been studied for other antibiotics that are also important and highly used, such as quinolones, macrolides, and diamino-pyrimidines.

Additionally, it is necessary to bear in mind that one of the potential sources of organic matter in water is that which is solubilized from the soil and ends up reaching waterbodies [19]. It also may be important to take into account the solubilized organic matter found in the soil solution, where there may also be antibiotics [20]. Soil organic matter is a complex mixture of different substances, the composition of which is not completely known due to the difficulty to accurately determining the molecular configuration of each compound, and also due to the fact that decomposition of organic matter in the soil by microorganisms takes place continuously [21]. Humic acids are the main components of soil colloidal material, which in turn represent between 70–80% of the total organic matter in soils [22,23], and are characterized by their high solubility at basic pHs and its insolubility at acid pHs, presenting molecular weight between 2000–10,000 [24].

Taking all this background into account, this study is aimed to evaluate the effects of pH and humic acids on the photodegradation of three different antibiotics. Specifically, they belong to three of the groups most widely used in human medicine, being chemically clearly different: Ciprofloxacin (which belongs to the group of quinolones), Clarithromycin (which belongs to the group of macrolides), and Trimethoprim (which is an antibiotic of the group of diaminopyrimidines).

## 2. Materials and Methods

### 2.1. Reagents

The antibiotics used were ciprofloxacin (Cip) (purity 98%), clarithromycin (Cla) (purity 98%), and trimethoprim (Tri) (purity 98%), which were supplied by Sigma Aldrich (Barcelona, Spain). Its main characteristics are presented in Table 1. The stock solutions for the antibiotics Cip and Tri were prepared in MilliQ water, and in the case of Cla it was prepared firstly at 5 mM in 96% Ethanol (Panreac, Madrid, Spain), and subsequently preforming a 50 µM solution with MilliQ water. The acetonitrile used for the analytical determination of antibiotics was HPLC grade and was supplied by Fisher Scientific (Madrid, Spain).

### 2.2. Photodegradation Experiments

The photodegradation experiments were carried out in a Suntest CPS^+^ simulator (Atlas, Chicago, IL, USA) equipped with a 550 W m^−2^ Xenon lamp, and with quartz filters, with a cut-off at 285 nm. It is characterized by having a light spectrum similar to that of natural sunlight, in which the temperature is maintained at 32 ± 2 °C. Exposure times to simulated light were in the range 0.0833–192 h. Simultaneously, another set of samples remained in the dark at the same time intervals. A 50 µM antibiotic solution was used, containing separately each of the three antibiotics. Six mL of solution were introduced into glass tubes (in triplicate), with some of them being exposed to simulated light, and other equivalent in number remained in the dark, all this during the different times of the experiments. The concentration of each of the antibiotics was analyzed by HPLC after the different contact times.

The equipment used for the quantification of antibiotics was an UltiMate 3000 HPLC chromatography device (Thermo Fisher Scientific, Madrid, Spain), with a quaternary pump, an autosampler, a thermostatted column compartment, and an UltiMate 3000 ultraviolet-visible detector. Attached to this equipment was a computer with version 7 of the Chromeleon software (Thermo Fisher Scientific, Madrid, Spain). Chromatographic separations were performed on a Luna C18 analytical column (150 mm long; 4.6 mm internal diameter; 5 μm particle size) from Phenomenex (Madrid, Spain), and a safety column (4 mm long; 3 mm ID; 5 μm particle size), packed with the same material as the column. The injection volume was 50 μL in the case of Cip and Tri, and 200 μL in the case of Cla. Flow rates were 1.5 mL min^−1^ for the first two antibiotics, and 1 mL min^−1^ for Cla. The temperature was kept constant at 25 °C throughout the analysis. Between each measurement, a wash was made with a solution composed of methanol and water (50:50).

The conditions in which Cla was separated were the following: the mobile phase consisted of acetonitrile (phase A), and 0.025 M monopotassium phosphate (phase B). The linear gradient elution program was run from 5 to 70% for phase A (and therefore 95 to 30% for phase B), with a time of 18 min. The initial conditions were restored in 2 min and were maintained for 3 min. The total analysis time was 25 min, with a retention of 13.6 min. The wavelength used for detection was 210 nm.

In the case of Cip and Tri, they were separated under the following conditions: the mobile phase was acetonitrile (phase A) and 0.01 M phosphoric acid (pH = 2) for phase B. The linear gradient elution program was executed from 5 to 32% for phase A and from 95 to 68% for phase B, in a period of 10.5 min. The initial conditions were restored in 2 min. The total analysis time was 15 min, with a retention time of 6.5 min for Cip and 5.6 min for Tri. The wavelength used was 212 nm.

After finishing with the quantification of all three antibiotics, a pseudo-first order kinetic model was used to describe kinetics results, as follows:*C⁄C*_0_ = *e^(−kt)^*(1)
where *C/C*_0_ is the fraction of the initial concentration (*C*_0_) that remains in the suspension after a given time *t* (expressed in h), and *k* (h^−1^) is the dissipation kinetic constant. The half-life (*t*_1/2_, expressed in h) of each compound was calculated as:*t(_1/2_) = ln 2/k*(2)

To study the photodegradation at different pH, it was firstly adjusted in the solutions corresponding to each antibiotic, to reach values of 4.0, 5.5 and 7.0, using 0.5 M NaOH or 0.5 M HCl. Subsequently, the same steps described above were followed until the completion of the quantification of each of the antibiotics.

The study of the effect of humic acids was carried out at an antibiotic concentration of 50 µM, with different concentrations of humic acids, specifically: 0.1, 0.2, 0.4, 1.0, 2.0, and 20.0 mg L^−1^. The pH of all solutions was adjusted to 5.5, with 0.5 M NaOH or 0.5 M HCl. The contact time was similar to the half-life of each of the antibiotics, being 15 min for Cip, 8 h for Tri, and 2 h for Cla. The half-lives were previously calculated from the experimental data of degradation in water at pH 4.0, described by the exponential decay model, from which the kinetic constant (*k*) was obtained, and from this the half-lives were calculated. The chemical characteristics of the humic acids used are described in [29].

Total organic carbon (TOC) was determined at time zero, 2 h, 16 h and 48 h (end of the experiment) by using a Multi N/C 2100 (Analytikjena, Jena, Germany). It was quantified for Cip and Tri samples. TOC was not measured in the Cla samples since ethanol was used when dissolving the antibiotic, as previously mentioned.

All determinations were made by triplicate.

## 3. Results and Discussion

### 3.1. Influence of pH

Figure 1, Figure 2 and Figure 3 show the results of the experiments corresponding to the degradation of Cip, Cla and Tri in milliQ water, at different pH values (4.0, 5.5 and 7.0), both with simulated light and in the dark.

None of the three antibiotics showed significant degradation in the dark, at any of the pH and contact times (between 0–72 h) tested. Belden et al. [30] also reported absence of degradation in the dark for Ciprofloxacin, and other researchers found the same result for other antibiotics, such as sulfonamides [16] and tetracyclines [17], as well as for pesticides such as carbofuran and metalaxyl [31]. However, in the current study the degradation of the antibiotics was evidenced when simulated light was used, and it took place following sequence: Cip > Cla > Tri.

Considering each antibiotic individually, starting with the one that showed the highest sensitivity to light, the degradation of Cip under simulated light was quite rapid for the three pH values studied, being above 85% after 1 h of exposure to the light radiation (Figure 1).

Cla degradation under simulated light was slower and decreased with increasing pH values. After 1 h of exposure, it reached 30% at pH 4.0, 27% at pH 5.5, and 17% at pH 7.0 (Figure 2).

Even slower was the degradation of Tri. After 1 h of exposure to simulated light, it reached less than 5%, at any of the three pH values tested (Figure 3).

The pseudo-first order kinetic model satisfactorily described experimental data, judging by the R^2^ values (ranging 0.906–0.990, Table 2). The half-life values followed the following sequence: Tri > Cla > Cip (Table 2), and generally decreased as a function of increasing pH, except for Cla. The kinetic constant *k* follows the reverse order, increasing with pH for both Cip and Tri, while it decreases for Cla (Table 2).

This behavior can be related to the percentages of the different species of antibiotics present as a function of pH, which are presented in Table 3, and which were calculated as per [32,33].

The distribution of species as a function of pH is similar for Cip and Tri, with dominance of positively charged forms at pH 4, while neutral forms predominate at pH 7. However, for Cla, positively charged species are predominant at both pH 4 and 7. Similar results were reported in a previous study [34], where the authors found that photodegradation of Cip increased as a function of rising pH, reaching a maximum at pH 8.6, which is related to its pK_a_ values (between 6.00–8.74, Table 1), indicating that the neutral form is more sensitive to photodegradation at slightly basic pH. At acidic pH, in which the COOH groups are not ionized, and the nitrogenous bases are fully protonated, Cip is more stable. Other authors also indicate that the degradation of Cip increases when pH raises in the range 4–7 [35,36]. These results may be due to the fact that the presence of hydroxide ions (OH−) can favor reaction with hydroxyl radicals generated by the presence of the antibiotic molecules in the solution, producing reactive oxygen species (O·−) [37]. The increase in pH also produces an enhanced degradation for other antibiotics, such as tetracyclines [17,38,39,40].

In addition, TOC was determined in each of the samples exposed to simulated light at different times and pHs (Table 4). The results indicate that the mineralization of Cip ranged between 27–37%, and that it was higher for Tri, reaching between 52–57%. Similar results had been obtained for other antibiotics, such as tetracyclines [17,40], with total mineralization of antibiotics hardly achieved, because the structure of aromatic rings tends to stabilize the oxidation of the by-products formed during photolysis.

### 3.2. Influence of Humic Acids

In general, the presence of humic acids did not cause any changes in the degradation of the three antibiotics when the experiments were carried out in the dark (Figure 4). Under simulated light, Cip degradation showed a slight increase when the highest concentration of humic acid was used, with values reaching 25%, while it was 16% in the dark (Figure 4a). In the case of Cla, no significant effects due to humic acids were observed as regards degradation (Figure 4b). However, for Tri, degradation showed a slight decrease under simulated light (Figure 4c), with *C/C*_0_ value going from 0.12 to 0.37 when the highest concentration of humic acids was added.

Previous studies reported that the presence of humic acids could cause opposite effects on the degradation of antibiotics, promoting it [16], or decreasing it [41]. This would be dependent on the concentration of such organic compounds, producing a balance between the photosensitizing effect of humic acids and the screen effect that occurs at high concentrations, in this last case preventing degradation [42]. This effect differing in function of the concentration of humic acids has been described for tetracyclines [39]. In the present work, the effect of humic acids was different depending on the antibiotic, with little or no effect being observed for Cip and Cla, while a decrease in degradation under simulated light was evidenced for Tri. Other studies indicated that the addition of dissolved organic matter does not have an effect on the degradation of Claritromycin exposed to light [43]. Finally, to be noted that Belden et al. [30] found that finely particulate organic matter initially accelerates the dissipation of ciprofloxacin, even in the dark, which may be due to the adsorption of the antibiotic in organic matter, an effect that has not been observed in the present work.

All these results could be useful to understand the probable evolution of these three antibiotics when they are spread in environmental comportments, and they are subsequently exposed to variable incidence of solar radiation, under natural conditions. In addition, the results could aid in case of programing eventual treatments applicable in WWTP, in order to achieve the photodegradation of these compounds.

Regarding future studies, this line of research could be further explored, with new antibiotics and with an additional variety of conditions tested. It would be interesting to shed further light and achieve further knowledge as regards eventual degradation under simulated natural sunlight, and also testing various catalyst materials with potential to enhance photocatalytic degradation.

## Figures and Tables

**Figure 1 molecules-26-03080-f001:**
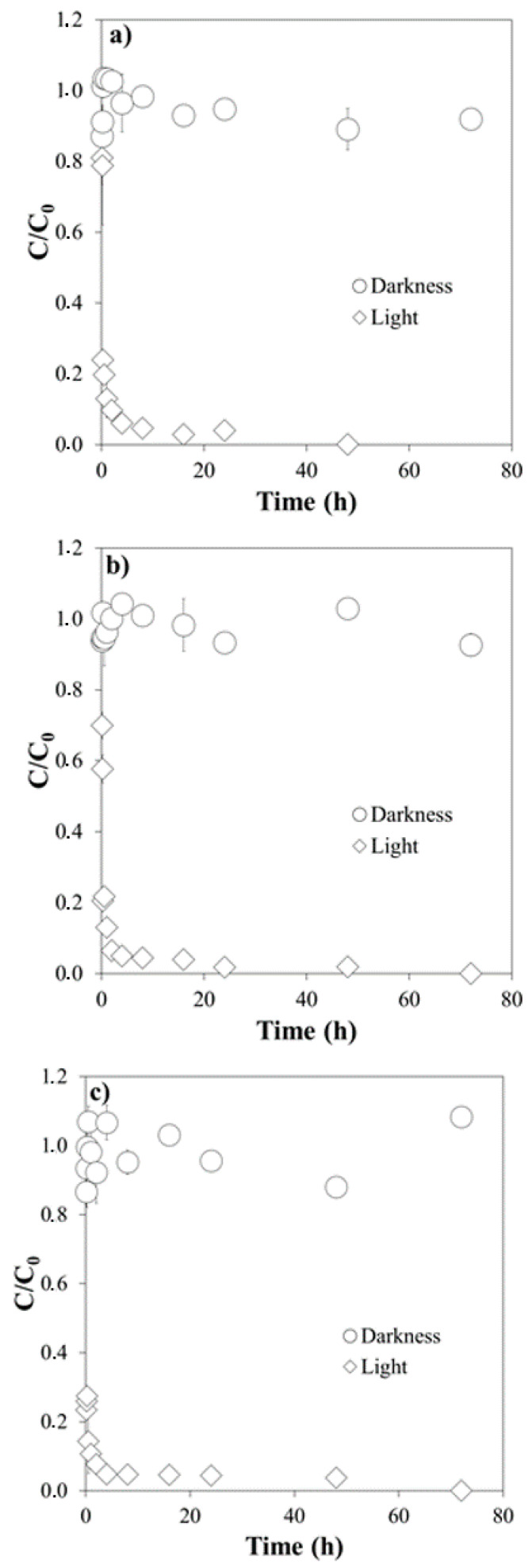
Degradation of Ciprofloxacin under simulated light and in the dark: (**a**) at pH 4.0; (**b**) at pH 5.5; (**c**) at pH 7.0. *C/C*_0_ is the fraction of the initial concentration (*C*_0_) that remains in the suspension after a given time t.

**Figure 2 molecules-26-03080-f002:**
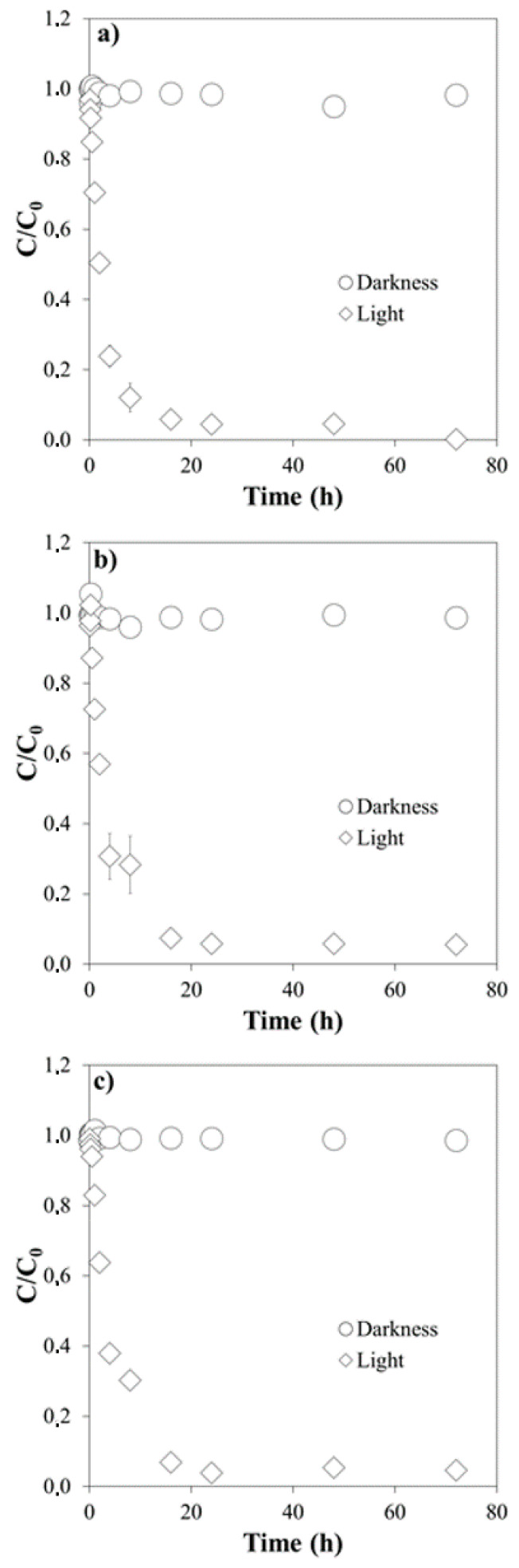
Degradation of Clarithromycin under simulated light and in the dark: (**a**) at pH 4.0; (**b**) at pH 5.5; (**c**) at pH 7.0. *C/C*_0_ is the fraction of the initial concentration (*C*_0_) that remains in the suspension after a given time t.

**Figure 3 molecules-26-03080-f003:**
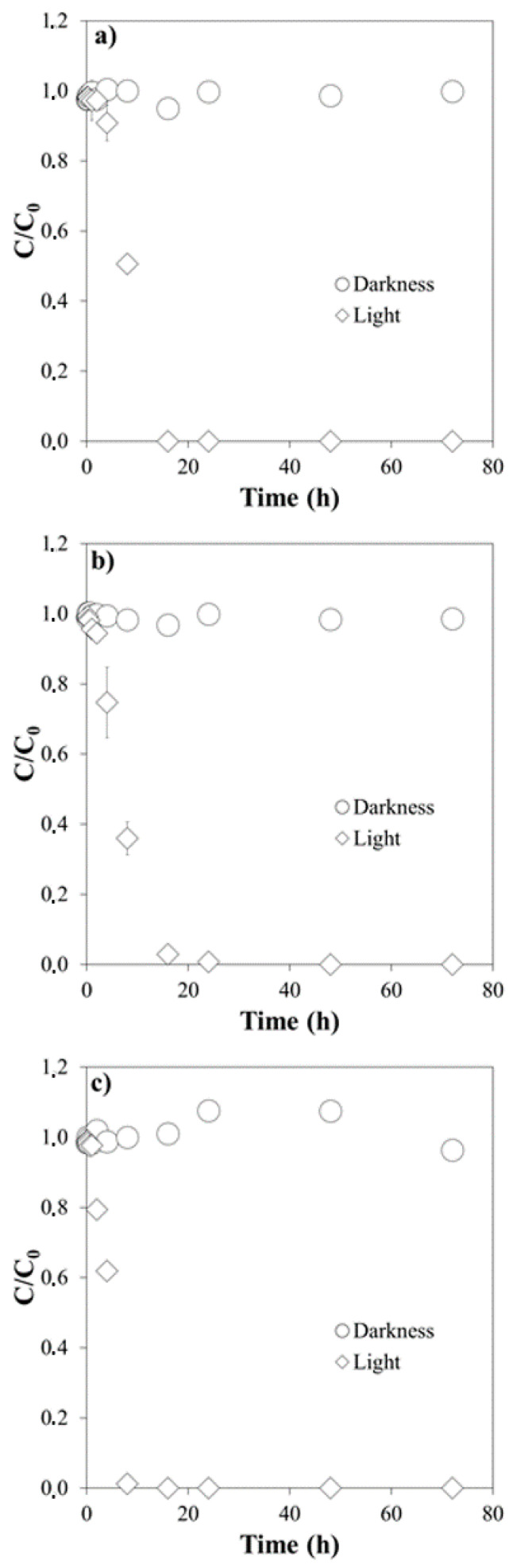
Degradation of Trimethoprim under simulated light and in the dark: (**a**) at pH 4.0; (**b**) at pH 5.5; (**c**) at pH 7.0. *C/C*_0_ is the fraction of the initial concentration (*C*_0_) that remains in the suspension after a given time t.

**Figure 4 molecules-26-03080-f004:**
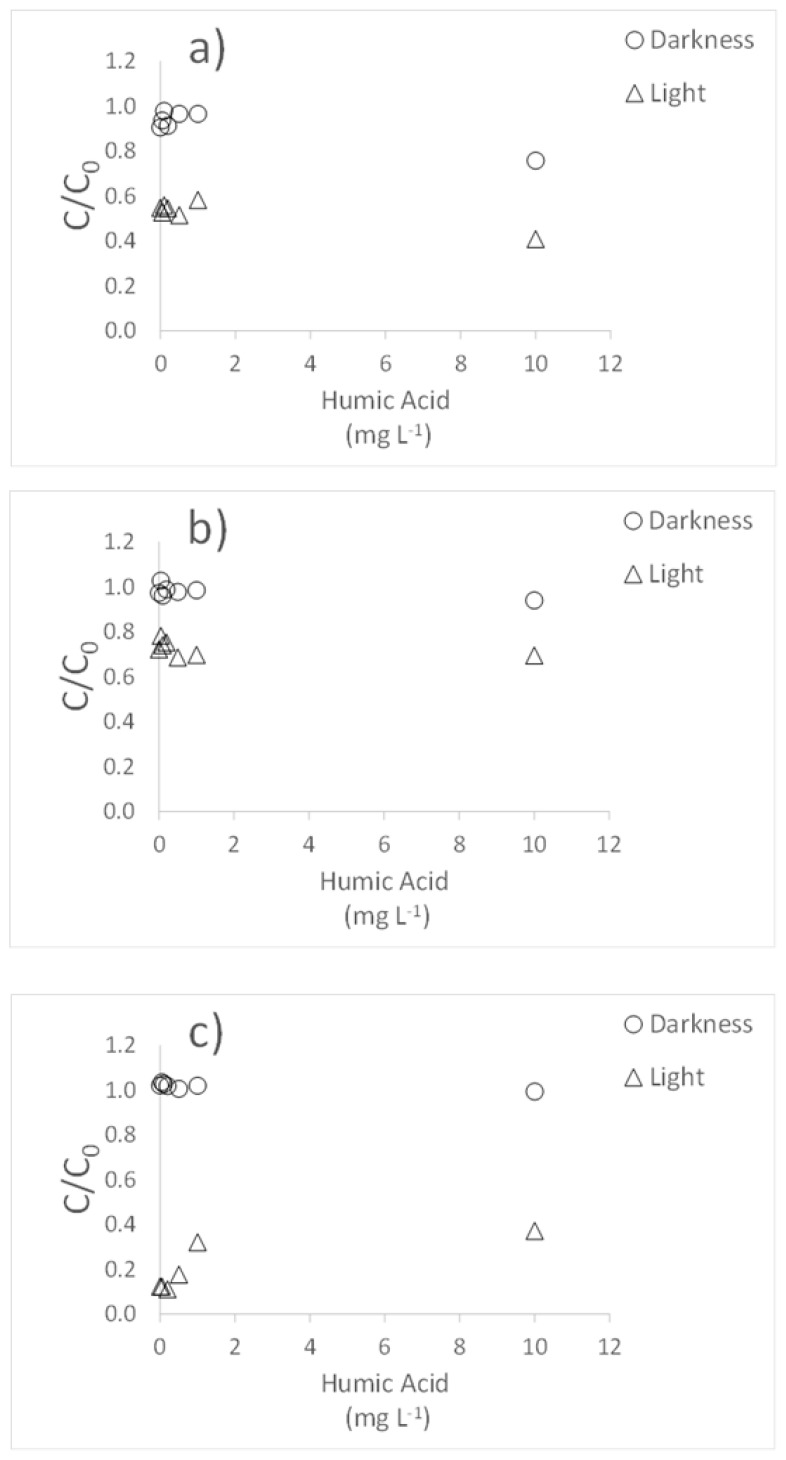
Influence of humic acids on the photodegradation of the three antibiotics: (**a**) Ciprofloxacin; (**b**) Clarithromycin; (**c**) Trimethoprim. *C/C*_0_ is the fraction of the initial concentration (*C*_0_) that remains in the suspension after a given time t.

**Table 1 molecules-26-03080-t001:** Main characteristics of the three antibiotics studied.

Common Name	Chemical Formula	Molecular Weight (g mol^−1^)	Log K_OW_ ^1^	pK_a_ ^2^	Water Solubility (mg L^−1^) ^1^
Ciprofloxacin ^1^	C_17_H_18_FN_3_O_3_	331.34	0.28	6.09–8.74	36,000
Clarithromycin ^2^	C_38_H_69_NO_13_	748.0	3.16	9.00–12.46	2
Trimethoprim ^3^	C_14_H_18_N_4_O_3_	290.32	0.91	6.16–7.16	400

K_OW_: n-Octanol/Water partition coefficient; ^1^ [25]; ^2^ [26,27]; ^3^ [26,28].

**Table 2 molecules-26-03080-t002:** Half-life (*t*_1/2_) and degradation kinetic constants (*k*) values for each antibiotic at the three pH tested. Average values (n = 3) ± standard error.

	*k* (h^−1^)	*t*_1/2_ (h)	R^2^
Ciprofloxacin			
pH 4.0	3.13 ± 0.48	0.22	0.966
pH 5.5	4.14 ± 0.47	0.17	0.981
pH 7.0	10.37 ± 2.07	0.07	0.906
Clarithromycin			
pH 4.0	0.34 ± 0.02	2.06	0.999
pH 5.5	0.25 ± 0.03	2.82	0.991
pH 7.0	0.19 ± 0.01	3.57	0.991
Trimethoprim			
pH 4.0	0.09 ± 0.02	7.35	0.978
pH 5.5	0.12 ± 0.02	5.81	0.989
pH 7.0	0.18 ± 0.03	3.95	0.984

**Table 3 molecules-26-03080-t003:** Percentages of different species of antibiotics as a function of pH. A: Antibiotic.

	A^+^	A^+/−^	A^−^
Ciprofloxacin			
pH 4.0	99.19	0.81	0.00
pH 5.5	79.54	20.45	0.01
pH 7.0	10.78	87.63	2.04
Clarithromycin			
pH 4.0	100.00	0.00	0.00
pH 5.5	99.97	0.03	0.00
pH 7.0	99.01	0.99	0.00
Trimethoprim			
pH 4.0	99.31	0.69	0.00
pH 5.5	81.73	17.88	0.39
pH 7.0	7.87	54.46	43.86

**Table 4 molecules-26-03080-t004:** Values of total organic carbon (TOC, mg L^−1^) at different times of exposure. to simulated light and at different pH values.

			Time		
	0 h	2 h	16 h	48 h	72 h
Ciprofloxacin					
pH 4.0	10.1	10.0	9.8	8.7	7.3
pH 5.5	10.1	9.4	8.0	7.7	7.3
pH 7.0	10.2	9.6	8.9	7.6	6.4
Trimethoprim					
pH 4.0	10.6	8.7	6.8	5.4	5.1
pH 5.5	10.7	9.0	5.9	5.5	4.9
pH 7.0	10.6	9.2	6.2	5.3	4.6

## Data Availability

Data is contained within the article.

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
