# Peer review of "Photodegradation of Ciprofloxacin, Clarithromycin and Trimethoprim: Influence of pH and Humic Acids"

_molecules, 2021, doi:10.3390/molecules26113080_

Round 1
Reviewer 1 Report
The research deals with the degradation proces of antybiotics. This is an area of research that has reached a certain maturity given the number of publications present in the current literature dealing with this topic. Given the previous state of the art of this area and the relative lack of novelty of the project, does not recommend an article for publication.
Author Response
Reviewer 1
The research deals with the degradation process of antibiotics. This is an area of research that has reached a certain maturity given the number of publications present in the current literature dealing with this topic. Given the previous state of the art of this area and the relative lack of novelty of the project, does not recommend an article for publication.
Response:
Thank you for your comments. In this regard, please take into account that searching on Scopus (using as keywords the name of each of the three antibiotics + photodegradation), the results are as follow:
Ciprofroxacin: 215 documents, from which 202 are articles. Including the word “pH” in addition to the search, just 60 documents remain (57 articles).
Clarithromycin: 19 documents, with none of them dealing with the influence of pH on degradation.
Trimethoprim: 0 documents.
In view of these results, we think that there are still aspects to investigate in this field.
Reviewer 2 Report
The article focuses on photodegration of three antibiotics, Ciprofloxacin, clarithromycin and Trimethoprim at three different pHs and in the presence of humic acid.
- The article has 2 takeaways: pH doesnt affect the concentration in the dark but causes degradation in the presence of light.
- Humic acid does not cause much effect.
The content in the article can be presented in much better way.
- Graphs 1(a-c), 2(a-c), 3(a-c) can be condensed to 3 graphs instead of 9 so that we can see how pH effects each antibiotic.
- The quality of the figures needs to be much better. There is no consistency in the quality of graphs 1,2,3 and figure 4.
- There is not sufficient explanation as to why degradation happens in the presence of light and not in darkness.
- What is the reason for studying the degradation of the antibiotics in humic acids? The motivation is not clear.
The content of the article needs to be overhauled majorly before acceptance.
Author Response
Reviewer 2
The article focuses on photodegration of three antibiotics, Ciprofloxacin, clarithromycin and Trimethoprim at three different pHs and in the presence of humic acid.
The article has 2 takeaways: pH doesn’t affect the concentration in the dark but causes degradation in the presence of light.
Humic acid does not cause much effect.
The content in the article can be presented in much better way.
Graphs 1(a-c), 2(a-c), 3(a-c) can be condensed to 3 graphs instead of 9 so that we can see how pH effects each antibiotic.
Response:
Thank you for your comments. Figures 1, 2 and 3 are now presented as Figure 1. We have also changed the X-axis to logarithmic scale in order to facilitate that the results can be more clearly seen and differentiated by the readers. We must also point out that, after having carried out tests, we have observed that grouping the graphs would lead to overlapping of various points, with which the results would not be clearly seen. It should also be noted that the old figure 4 now becomes figure 2.
The quality of the figures needs to be much better. There is no consistency in the quality of graphs 1,2,3 and figure 4.
Response:
Thank you for your comment. We have improved it.
There is not sufficient explanation as to why degradation happens in the presence of light and not in darkness.
Response:
Thank you for your comment. We have further clarified it in the new version of the manuscript, both in the Introduction and in the Results and Discussion sections.
What is the reason for studying the degradation of the antibiotics in humic acids? The motivation is not clear.
Response:
Thank you for your question. This aspect has been clarified in the Introduction section of the revised manuscript.
The content of the article needs to be overhauled majorly before acceptance.
Response:
Thank you for your indication. The manuscript has been revised and corrected in order to clearly improve it.
Reviewer 3 Report
In the manuscript, data related to the photodegradation of three antibiotics, namely, Ciprofloxacin (Cip), Clarithromycin (Cla) and Trimethoprim (Tri), at three different pH (4, 5.5 and 7) and in the presence of different concentrations of humic acids are shown.
In my opinion, I found this article more like a general report lacking both explanations and proper graphical presentations. I would strongly recommend revision before its publication. I have the following suggestions for the authors;
- As the central core of the paper is associated with the photodegradation of the three antibiotics with a light source that mimics the solar spectrum, it would be an evident curiosity of the reader to know the absorption profile of each antibiotic studied in the present case. Also, the effect of pH on the absorption profile should be given. A spectrum of the light source also can be shown. Ultimately, a brief mechanism/steps involved in photodegradation should be included.
- All plots are needed to improve both to represent information and quality. For example, 2-i. Fig. 1 – 3, either plot those data in ln (C/C0) v/s Time or use two Y-axis (say left side for dark, right side for light) if the authors don’t wish to use ln scale. 2-ii These plots might look much better if the range for X-axis was 0 – 20 (h). 2.-iii. Similarly, fig. 4 should be improved.
- An insight or some information on the photoproducts should be provided.
- Check spellings/English; few examples
P12-L249: comportments à compartments
P12-L252: WWLP ???
In Ref 36, instead of high, the authors used hight.
Author Response
Reviewer 3
In the manuscript, data related to the photodegradation of three antibiotics, namely, Ciprofloxacin (Cip), Clarithromycin (Cla) and Trimethoprim (Tri), at three different pH (4, 5.5 and 7) and in the presence of different concentrations of humic acids are shown.
In my opinion, I found this article more like a general report lacking both explanations and proper graphical presentations. I would strongly recommend revision before its publication. I have the following suggestions for the authors;
As the central core of the paper is associated with the photodegradation of the three antibiotics with a light source that mimics the solar spectrum, it would be an evident curiosity of the reader to know the absorption profile of each antibiotic studied in the present case. Also, the effect of pH on the absorption profile should be given. A spectrum of the light source also can be shown. Ultimately, a brief mechanism/steps involved in photodegradation should be included.
Response:
Thank you for your comment. Most of that information is not available at this moment, but we agree with the Reviewer in considering it interesting, thus meriting specific research, and we will program further complementary experiments in this regard for the coming future, which would aid to elucidate these aspects. In addition, in the revised version of the manuscript, we have included more details regarding the lamp used as light source.
All plots are needed to improve both to represent information and quality. For example, 2-i. Fig. 1 – 3, either plot those data in ln (C/C0) v/s Time or use two Y-axis (say left side for dark, right side for light) if the authors don’t wish to use ln scale. 2-ii These plots might look much better if the range for X-axis was 0 – 20 (h). 2.-iii. Similarly, fig. 4 should be improved.
Response:
Thank you for your comment. The figures have been modified, and the logarithmic scale has been used in the X axis.
An insight or some information on the photoproducts should be provided.
Response:
Thank you for your comment. In the current work we did not study in detail metabolites resulting from photodegradation, but it is one of the aspects that we would consider in future complementary research. Any case, we include some information as regards metabolites derived from Ciprofloxacin, which is the antibiotic that has been most studied among the three here considered.
Check spellings/English; few examples
P12-L249: comportments à compartments
P12-L252: WWLP ???
In Ref 36, instead of high, the authors used hight.
Response:
Thank you for your comment. We have checked it and corrected all typos and mistakes detected.
Reviewer 4 Report
In this manuscript, the authors reported the effects of pH and humic acids on the photodegradation of three types of antibiotics Ciprofloxacin, Clarithromycin and Trimethoprim under simulated light irradiation. This manuscript is somewhat interesting in the removal of environmental pollutants. In my opinion, this manuscript may be considered for publication in Molecular after minor revision.
(1) In the Introduction, the authors should also mention the photocatalytic degradation of antibiotics and other pollutants since photocatalysis has been received as an important green technology to degrade various pollutants including antibiotics (e.g. Adv. Powder Technol. 32 (2021) 951–962, J. Electron. Mater. 49 (2020) 5248–5259).
(2) The authors investigated the effect of pH = 4, 5.5 and 7 on the photodegradation of antibiotics, why did not they investigate the effect of pH > 7?
(3) Please describe and explain the data present in Table 3 so that readers can understand them more easily.
(4) The authors are suggested to analyze the photodegradation mechanism of antibiotics more detailedly, especially the photodegradation difference between the three types of antibiotics at different pH values. This could attract more attention from readers.
Author Response
Reviewer 4
In this manuscript, the authors reported the effects of pH and humic acids on the photodegradation of three types of antibiotics Ciprofloxacin, Clarithromycin and Trimethoprim under simulated light irradiation. This manuscript is somewhat interesting in the removal of environmental pollutants. In my opinion, this manuscript may be considered for publication in Molecular after minor revision.
(1) In the Introduction, the authors should also mention the photocatalytic degradation of antibiotics and other pollutants since photocatalysis has been received as an important green technology to degrade various pollutants including antibiotics (e.g. Adv. Powder Technol. 32 (2021) 951–962, J. Electron. Mater. 49 (2020) 5248–5259).
Response:
Thank you for your comment. We have mentioned photo-catalysis in the Introduction section of the revised version of the manuscript, and we have incorporated the references suggested by the Reviewer.
(2) The authors investigated the effect of pH = 4, 5.5 and 7 on the photodegradation of antibiotics, why did not they investigate the effect of pH > 7?
Response:
Thank you for your comment. The pH values of the soil solution are in most cases below 7, and it is also frequent in wastewater. However, it could be also interesting considering pH values above 7 in further complementary future experiments.
(3) Please describe and explain the data present in Table 3 so that readers can understand them more easily.
Response:
Thank you for your comment. We have further explained it to facilitate understanding of the information included.
(4) The authors are suggested to analyze the photodegradation mechanism of antibiotics more detailedly, especially the photodegradation difference between the three types of antibiotics at different pH values. This could attract more attention from readers.
Response:
Thank you for your comment. We have modified the paragraph corresponding to photodegradation, trying to improve it and increase the interest, as well as facilitate understanding as regards this key aspect.
Round 2
Reviewer 3 Report
Dear Editor,
The authors have revised the manuscript considerably. However, some areas are still not covered in the revised manuscript, but they explained the reasons behind it. I believe now ms can be accepted in its present form.
Thank you,
Neeraj
Author Response
Comments and Suggestions for Authors
Dear Editor,
The authors have revised the manuscript considerably. However, some areas are still not covered in the revised manuscript, but they explained the reasons behind it. I believe now ms can be accepted in its present form.
RESPONSE: Thank you for your comment and indication that the manuscript can now be accepted in its present form.
In view of that, no further modification would be carried in the current manuscript.